# Healthy-Sustainable Housing Index: A Pilot Study to Link Architecture and Public Health in a Semi-Urban Community in Mexico

**DOI:** 10.3390/ijerph16030295

**Published:** 2019-01-22

**Authors:** Pamela Zúñiga-Bello, Astrid Schilmann, Eunice Félix-Arellano, Gerardo Gama-Hernández, Urinda Alamo-Hernández

**Affiliations:** 1Environmental Health Department, Center for Population Health Research, National Institute of Public Health, Av. Universidad 655, Col. Santa Ma. Ahuacatitlán, Cuernavaca 62100, Mexico; pamela.zuniga@espm.insp.mx (P.Z.-B.); aschilmann@insp.mx (A.S.); eunice.feliz@espm.insp.mx (E.F.-A.); 2Urbanism academy, Faculty of Architecture, Autonomous University of Morelos, Av. Universidad 1001, Col. Chamilpa, Cuernavaca 62209, Mexico; ggama@uaem.mx

**Keywords:** healthy-sustainable housing index, respiratory symptoms, pilot study, Mexico

## Abstract

The aim of this pilot study was to evaluate the link between housing and children´s respiratory symptoms, through the construction of an index (HSHI) based on the definition of healthy-sustainable housing criteria, in a semi-urban community from Morelos, Mexico. A general and household questionnaire, and respiratory symptoms diary were applied in 60 households to gather information about schoolchildren, respiratory health, housing and lifestyle characteristics. HSHI was constructed using principal component analysis. The association between HSHI and the presence and duration of respiratory symptoms was assessed using logistic and Poisson regression models. HSHI had five components, which accounted for 63% of variance, and were classified into poor and sufficient quality. It was observed that schoolchildren who inhabit a sufficient-quality house, showed a reduction in nose irritation duration and in the allergic symptoms probability regarding component 1 (ventilation, lighting and cloth washing) and presented three times less duration of common cold by component 2 (construction material, painted walls inside the house and type of bathroom) compared to poor-quality house inhabitants. Our results suggest that living in a sufficient-quality house, as described by the HSHI, reduced the prevalence of wheezing episodes and the probability of ear pain, providing evidence about the positive association of a healthy-sustainable housing on the respiratory health of schoolchildren.

## 1. Introduction

Housing is considered as the physical structure and psychosocial environment used for habitation [1]. A healthy housing consists of the physical and psychosocial environment that prevent risk factors and could impact the health or welfare of their inhabitants [1]. It is also defined as the site designed, built, renovated, and maintained in ways that support the health of residents [2]. The criteria to define a healthy housing, according to the Pan American Health Organization (PAHO) and the World Health Organization (WHO) include: safe location and secure tenure, adequate design and structure, space for a healthy coexistence, basic facilities, appropriate environment, healthy habits and protection against adverse health effects [1,3]. Engineers and architects play a major role in the implementation of these criteria during design and construction and should be aware of their impact on the welfare, health, and quality of life of the inhabitants [4,5].

Architecture embraces a wide range of styles in a constant evolution toward sustainability. Sustainable architecture is within an ideal ecological, social and economic context to create and operate a healthy built environment [5,6,7,8]. It includes the use of alternative sources of energy, environmental management through bioclimatic design, and a development that meets the needs of the present generation without compromising the ability of future generations to meet their own [9,10]. Moreover, sustainable architecture argues that the quality of the indoor environment can promote healthy housing and an improvement in the health of its inhabitants [4,5]. 

Despite the emergence of these architectural styles, the complex relationship between "sustainable housing" and health is just beginning to be studied [4,11,12,13,14,15]. Sick building syndrome [16,17,18,19] and housing have been associated with an increased risk of accidents and adverse health effects [1,12,16,20,21,22,23] like respiratory diseases. Acute respiratory infections (ARIs) can be attributed to several risk factors such as the environment, and housing characteristics fall within the latter [21,24,25]. At the global level, studies regarding the housing-health relation are mostly focused on respiratory diseases [11,12,16,17,18,19,20,25,26,27,28,29,30]. Nevertheless, ARIs imply a public health problem in México. In Alpuyeca, Morelos, ARIs are the first leading cause of medical consultation (13.04 cases per 1000 inhabitants) [24,31,32]. 

Health impacts of selected housing risk factors have also been quantified applying the environmental burden of disease approach in Europe [3]. In addition, the economic implications of inadequate housing have been studied through a Housing health and Safety Rating System (HHSRS) in England [3]. The HHSRS includes 29 potential hazards classified as: physiological requirements (e.g., damp and mould, excessive cold, lead, radiation, volatile organic compounds), psychological requirements (e.g., lighting, noise, crowding and space, entry by intruders), protection against infection (e.g., domestic hygiene, drainage), and protection against accidents (e.g., electrical hazards, structural collapse and falling elements) [3]. A National Healthy Housing Standard has been proposed in USA as a tool to reconnect the housing and public health sectors, including the minimal standards for a safe and healthy home [33]. However, many of the previously described standards are far from the Mexican and Latin American reality.

On the other hand, various authors have proposed the construction of indexes to provide a measure of the ‘healthiness’ and ‘safety’ of housing [26] that include housing characteristics related to the health of its inhabitants [34,35,36,37,38]. In Mexico, the Mario Molina Center proposed a housing and environmental sustainability index (Sustainability Index of Housing and its Surroundings, Índice de Sustentabilidad de la Vivienda y su Entorno, ISV in Spanish) [39]; however, this index does not include the variable of health of its occupants. Despite these efforts, there is a need for further research in this emerging area [4,14,16].

In this regard and looking to build a bridge between architecture and public health, the aim of this pilot study was to evaluate the link between housing and children´s respiratory symptoms, through the construction of a housing index (HSHI) based on the definition of healthy-sustainable housing criteria (Figure 1), in the semi-urban community of Alpuyeca Morelos, Mexico. This study is part of a project called Centro Asociado en Salud Infantil y Tópicos Ambientales (CASITA) [40].

## 2. Materials and Methods 

### 2.1. Study Population 

A cross-sectional pilot study was conducted from November 2014 to February 2015, in schoolchildren between 7 and 12 years old, who were residents of Alpuyeca for at least three years. Participants were selected based on a random sampling stratified by gender and school grades of elementary schools (as part of the CASITA project). Parents of the selected children were invited to participate in the study, and those who accepted the home visit and answered the housing questionnaire composed the final sample of 60 children. The project was approved by the Ethics, Biosecurity and Research committees of the National Institute of Public Health in Mexico (Instituto Nacional de Salud Pública INSP in Spanish). 

### 2.2. Instruments for Collecting Information

The following instruments were applied:

(1) General Questionnaire: Applied to the mothers of participating schoolchildren by 3 trained students of the Master of public health at the INSP. It included sociodemographic variables, housing characteristics, lifestyle, and anthropometric measurements.

(2) Housing Questionnaire: Applied during the home visit to an adult inhabitant of the house by an architect studying the Master of public health at the INSP (Pamela Zuñiga Bello PZB). It included variables such as the type of boundaries, hygiene habits, and use of cleaning products, maintenance, use of pesticides and fuels in the house, pets, and the presence of mildew and dampness. The following measurements were conducted in the room of the schoolchildren: height and area of the room, as well as the percentage of lighting and ventilation of the space. Ventilation and lighting were defined according to the building code from Cuernavaca municipality in the state of Morelos. This regulation considers the room area and the windows dimension to define an adequate ventilation (windows must cover at least the 10% of the room surface). The natural lighting from windows should cover at least the following percentage according to its location: north 10%, south 15%, east 12% and west 11% [41].

(3) Respiratory symptoms log: A four-week diary was designed to record the presence or absence of the respiratory symptoms listed below. The recruitment of participants was carried out during the period from November to December. Mothers were trained to enter the information in this diary, from which the following respiratory profiles were defined: (a) Acute upper respiratory infections (URI). Presence of at least two of the following symptoms: fever, nasal obstruction or runny nose, dry cough or coughing up mucus and sore throat [24,37,42].(b) Acute lower respiratory infections (LRI). Presence of rapid breathing and cough or shortness of breath [43].(c) Respiratory allergy. Presence of sneezing, eye irritation or burning in the nose [44].

The number of episodes and the incidence of these respiratory profiles were calculated. A new episode was counted as the presence of a respiratory symptom after the child has been symptom-free for at least one day [45], while the incidence was defined as new episodes/time at risk [46]. 

(4) Participant Observation: Performed during the informal visit by the architect who applied the housing questionnaire and two trained housing inspectors (construction technician students). Qualitative notes regarding the housing characteristics and cleaning habits were taken.

(5) Informal Interviews: Intended to supplement the information gathered in the general and housing questionnaires during their application. 

### 2.3. Healthy-Sustainable Housing Index (HSHI)

Twenty-six variables were selected to be included as HSHI criteria (Table 1) in accordance with national and international healthy housing criteria [1,8,47,48,49,50] of sustainable architecture (Figure 1), and finally adapted to the local context in Alpuyeca. HSHI was constructed using principal component analysis (PCA) [51,52] on the correlation matrix. Components were selected according to their eigenvalue >1 and considering the variability explained by each of the components and collectively [51,52]. To make up each component, dominant variables were identified, i.e., those that presented an eigenvector greater than an established cut-off point in ±0.30 (Table 2). Once the components were selected, the score for each one was generated. Components were categorized as dichotomous variables [51] where the 50th percentile (median), for the continuous score, was set as the cut-off point to indicate poor or sufficient housing quality for that component. The global score of the HSHI was obtained by adding the scores of each component so that houses were classified within a category of poor (6 points) or sufficient quality (14 points). 

### 2.4. Statistical Analysis 

The study population was described through measures of central tendency and dispersion, as well as the prevalence and incidence of symptomatology. The association between the frequency of respiratory symptoms and other relevant covariates, different to housing, was assessed using the X^2^ test and Fisher exact test. Consequently, and through these same tests, the relationship between components/HSHI and the presence of respiratory symptoms was assessed. To estimate the association between the components/HSHI and the presence and duration of respiratory symptoms and/or profiles, logistic and Poisson regression models were constructed respectively. Multivariate models were used and adjusted by the characteristics that were significant in the bivariate analysis and by those identified in the literature [51,52,53,54]. The STATA program version 13 (Stata Corp LP, College Station, TX, USA) was used for the analysis. 

## 3. Results

### 3.1. Characteristics of the Houses Surveyed. 

Of the 60 houses surveyed, five (8.33%) were not included in the study due to the lack of information on the monitoring of schoolchildren respiratory symptoms. More than half of the participating houses have adequate ventilation and lighting. Around the same percentage reported to have less than six inhabitants, the presence of pests, and cleaning the house at least once a week. The presence of dampness and mildew was observed in approximately half of the houses (Table 1). A similar proportion of boys and girls participated in the study, with an average age of ~10 years. Also, most of the parents reported an elementary education level (Table 2).

### 3.2. Construction of the Healthy-Sustainable Housing Index (HSHI)

Table 3 shows the five selected components representing 63.38 % of the original variance. Of the 60 houses selected, 53% were located in the category of poor housing quality and 47% in the category of sufficient housing quality. The HSHI indicates that the houses selected are, on average, in the medium-high category, reaching the range of 6 to 14 points (median 9.88 ± 1.88) on a scale from 0 to 14.

### 3.3. Respiratory Symptoms and HSHI 

It was observed that schoolchildren presented at least 1 episode of a respiratory symptom and/or profile during the follow-up period. Among the recorded symptoms, dry cough and common cold presented the highest prevalence, while LRI and wheezing presented the lowest, although no statistically significant differences were observed (Figure 2a). It was observed an incidence of ≥10 episodes/child-year in the symptoms of runny nose, sore throat, common cold and sneezing. On the other hand, wheezing and LRI reported the lowest number of episodes (less than 2 episodes/child-year) (Figure 2b). 

In houses with poor-quality, it was observed that about half of the mothers or guardians of schoolchildren reported that their children had at least dry cough and runny nose on one occasion, while a quarter presented URI and a fifth part showed allergy on at least one occasion during the follow-up period (Figure 2b). When comparing the respiratory outcomes between the houses with a poor and sufficient quality, only significant differences (*p* < 0.05) were observed in the presence of wheezing. However, there is a trend for more frequent respiratory symptoms in houses with poor- quality (Figure 2b). 

Table 4 shows the crude and adjusted odds ratio (OR) from the logistic regression models of the prevalence and profiles of respiratory symptoms for the HSHI, and each one of the 5 components. The presence of dry cough, runny nose, ear pain, and hoarseness, itching in the nose, URI, and allergy were associated with the first component (ventilation, lighting and washing work and family clothing together). This indicates that schoolchildren who live in a poor-quality house, have 5.32 times more probability of presenting allergic symptoms compared with those who live in a sufficient-quality house described by component 1, without adjusting for other covariates. Also, it was observed that the schoolchildren who live in a house with a poor-quality have 5.32 times more probability of suffering from ear pain compared with those who live in a house with a sufficient-quality HSHI, however, this association was marginally significant in the adjusted model. Contrary to what was expected, schoolchildren who inhabit a house of poor-quality have 11.63 times less probability of presenting common cold, compared with those who inhabit a house of sufficient-quality described by component 2 (Table 4).

The Poisson regression models also show an association between respiratory symptoms duration and the first component. The adjusted analysis shows that schoolchildren who inhabit a sufficient-quality house had 4 times shorter duration of nose irritation compared to schoolchildren who inhabit a poor-quality house regarding component 1. On the other hand, duration of sneezing in schoolchildren who inhabit a sufficient-quality house for component 3 was 1.8 times larger compared with those who inhabit a poor-quality house. The multivariate analysis highlights an association between component 2 and common cold and URI, where schoolchildren living in a sufficient-quality house described by this component, present a 2.9 and 4.3-times less duration compared with schoolchildren in a poor-quality house. Likewise, the association between component 5 and sore throat indicates that schoolchildren who live in a sufficient-quality house, have a 2.7 times larger duration of sore throat compared with those who live in a poor-quality house described by the same component (Table 4).

## 4. Discussion

The main findings of this pilot study suggest that living in a house with a sufficient-quality, expressed by the HSHI, lowered the frequency of some of the respiratory symptoms, which is evidenced by the reduction of the prevalence of wheezing episodes and of the probability of suffering from ear pain. In addition, the housing quality assessed by the component 1 reduced the presence and duration of symptoms such as allergy, nose irritation, runny nose, and URI, while component 2 was associated with a lower probability of suffering common cold and URI; components 3 and 5 were also identified as risk factors for sneezing, sore throat, and URI. These findings were statistically significant.

To the best of our knowledge, this is the first pilot study that evaluates the respiratory health and the quality of the housing through a "healthy-sustainable housing index" in Mexico. Our results for component 1 are consistent with those reported by Fisk et al; 1997 who noted that a high-performance ventilation system reduces respiratory diseases by 9–20% [55]; while Norhidayah et al; 2013 suggest that ventilation is one of the significant predictors of diseases related to buildings [19]. In turn, Hesselmar et al; 2005 reported that inadequate ventilation is associated with the presence of wheezing (OR = 3.13, IC 1.05–9.29) and asthma (OR = 1.1, IC 0.48–2.48) [54]. On the other hand, the construction of component 2 included variables such as construction material (living room ceiling), indoor painted walls and type of bathroom, features that were deemed fundamental for establishing a housing-health relationship [53,54,55,56,57]. In component 3, characterized by the construction material of the house’s floor, it was noted that 29.63% of houses with floors made of earth/wood or other coatings, showed better hygiene habits that those with concrete or surfaced floor (results not displayed in tables). Regarding component 5, which considers the drinking water supply inside the house, it was noted that, despite the fact that most of the population (68.52%) had this service, it is only available once a week in 59.26% of households, which could influence the presence of moisture (52.73%) and mold (43.64%) given the storage practices. The foregoing is consistent with Cuijpers et al; 1995 and Hagmolen et al; 2007 whose results indicate that moisture has been associated with respiratory symptoms such as wheezing (OR_BOYS_ = 1.86, IC 0.22–1.44, or OR_GIRLS_ = 1.48 IC 0.62–3.54), while the mold and moisture located in the living room or children rooms, are associated with respiratory diseases such as asthma (OR = 3.95, IC 1.82–8.57) [53,54].

In addition, the presence of respiratory symptoms reported in this pilot study may have been influenced by the geographical and sociocultural context [21,57]. Since it was conducted in a semi-urban area, the quality of the indoor/house environment might not be a priority because of the urgency to address other economic and social problems [21,57,58]. In turn, according to other studies, there is widespread ignorance about the relationship between housing and respiratory health [4,12], which is consistent with the information gathered in the informal interviews. It is possible that the increase in the frequency of some episodes in schoolchildren living in a sufficient-quality house, could be influenced by a higher socio-economic status since they have greater possibilities to visit a physician to detect the symptoms.

When interpreting our results, the methodological strengths and limitations shall be considered. The sample size (*n* = 55) limited the associations found between the HSHI and the respiratory symptoms, even with the results found when performing the PCA. Regarding the measurement of the symptoms, the present pilot study does not have a clinical diagnosis of symptoms/respiratory profiles, so that the construction of the tracking log was based on other studies [13,25,28,30]; therefore, we cannot rule out a measurement error. During the study period, mothers were reminded on the follow-up of the symptomatology. However, it cannot be ruled out that there is a memory bias. To reduce the bias of information regarding the data of the housing, the visits were carried out by technicians trained in construction who were unaware of the hypothesis of the pilot study.

Certain characteristics deemed important in previous studies could not be included in our analysis due to the little-attributed variability, such is the case of living in multifamily dwelling [37,58], the age and remodeling of the construction, homeownership [29], as well as having a comfortable temperature in the interior of the house, and the type of construction material used in the roof of the schoolchildren’s bedroom [36]. Likewise, relevant data regarding exposure were not considered which, according to the literature, would have made it possible to establish associations with greater certainty, such as relative humidity, indoor and outdoor ambient temperature, indoor samples of mold and dust, the number of hours that mothers keep the windows open or the number of hours that schoolchildren spend inside the house [26,27,30,36,37,59,60,61]. Moreover, it was not possible to determine whether schoolchildren had a deficiency of vitamin A, a characteristic that has been associated with the presence of URI [42,62]. Also, the high incidence of symptoms such as dry cough, runny nose, sore throat, and sneezing, could be related to the follow-up period (November 2014–February 2015). However, there is no seasonality adjustment, since climate change is one of the major factors increasing the incidence of respiratory diseases [29,56]. 

While it is true that housing in Alpuyeca was placed within a medium-high quality, it should be noted that the construction of the HSHI is based on the current characteristics of the housing of this community and not on the ideal healthy [1,4] and sustainable criteria [1,2,3,4,5,6,7,8,47,48,49,50].

## 5. Conclusions

The results evidence the relationship of the healthy-sustainable housing and the respiratory health of its inhabitants through the proposed HSHI. In Mexico, the promotion of sustainable housing is scarce in spite of having a national criterion. While the efforts to promote healthy housing have been greater, there are no specific criteria for the country, which draws on the recommendations suggested by the Pan American Health Organization/World Health Organization (PAHO/WHO). At our discretion and according to other studies, the transdisciplinary work between the community, specialists in public health, environmental health, architects, engineers and other stakeholders, is necessary to encourage the efforts to promote a healthy-sustainable housing and benefit the respiratory health and quality of life of the Mexican population. These efforts should start with the inclusion of this knowledge in the professional education of all key elements.

This pilot study constitutes a platform for raising new research on this topic. Future projects should consider higher sample sizes, longer follow-up periods, different seasons of the year, comparison groups regarding the type of housing, other public health problems in addition to respiratory health, and a qualitative-quantitative methodology.

## Figures and Tables

**Figure 1 ijerph-16-00295-f001:**
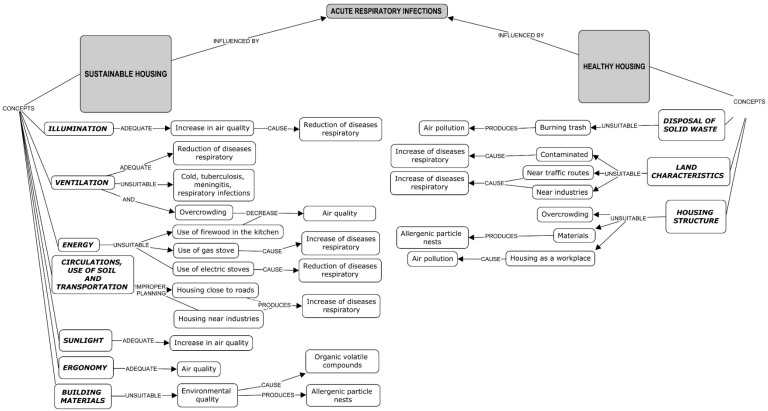
Conceptual framework considering the relationship between acute respiratory infections, sustainable housing, and healthy housing. Source: Own creation [1,4,5,6,7,8,9,11,12,13,22,34,35,36,37,39].

**Figure 2 ijerph-16-00295-f002:**
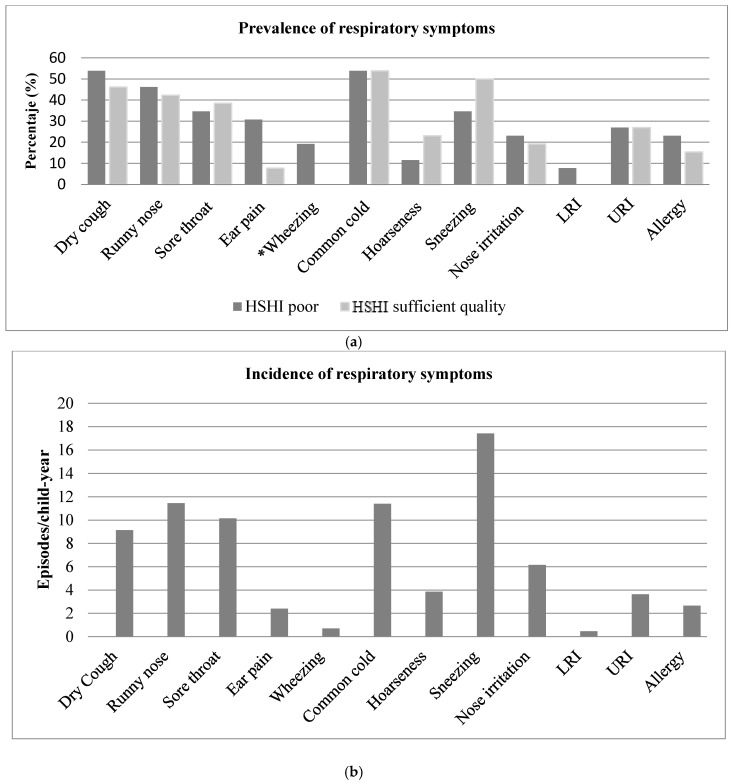
Frequency of respiratory symptoms for children during the follow-up period in Alpuyeca Morelos. November 2014–February 2015. *N* = 55. (**a**) Incidence (episodes/child-year); (**b**) Prevalence regarding HSHI. ***** Difference of proportions between the HSHI categories statistically significant (*p* < 0.05). Schoolchildren showed some symptom at least once during the month follow-up period.

**Table 1 ijerph-16-00295-t001:** Description of the characteristics of the selected houses in Alpuyeca, Morelos.

Number of Houses (55)		
Characteristic	*N*	%
Area where house is located ^a^		
House located on paved street	34	61.8
House located on gully/bank of the river/channel	38	69.0
One story house ^a^	45	81.8
Suitable height (>2.40 m) ^a^	51	92.7
Homeownership ^a,b^	35	66.0
Age of construction >10 years	27	49.0
Adequate ventilation ^a^ (according to local building code)	32	58.1
Adequate lighting ^a^ (according to local building code)	21	61.8
Period living in the house >10 years ^a^	17	30.9
Parental presence in the household (Mom, dad and children/ Dad or mom with children) ^a^	35	63.6
Number of people living in the house (< 6 people)	37	63.6
Multi-family dwelling (2 to 50 houses) ^a^	31	56.3
Possession of pets (cat, dog, birds)	46	83.6
Presence of pests (rats, mice, cockroaches)	28	53.8
Cleaning of the house less than once a week ^a^	32	58.1
Cleaning product		
Use of chlorine as cleaning product	20	36.3
Use of floor cleaners/ glass cleaner / aromatizer as cleaning product	35	63.6
Presence of dampness	29	52.7
Presence of Mold	24	43.6
The house has been waterproofed in the last 5 years ^a^	14	25.4
The house has been remodeled in the last 6 years ^a^	17	30.9
Floor (the house has at least a space with this type of material) ^a,b^		
Earth/ wood or other coatings	16	29.6
Concrete/or surfaced floor	38	70.3
Living room ceiling (the roof is built with any of the following materials) ^a,b^		
Cardboard, palm, shingle (tejamanil) or wood, metal foil, foil of asbestos, does not know.	26	48.1
Concrete slab, bulkhead, brick, block	28	51.8
Kitchen ceiling (the roof is built with any of the following materials) ^a,b^		
Cardboard, palm, shingle (tejamanil) or wood, metal foil, foil of asbestos, does not know.	29	53.7
Concrete slab, bulkhead, brick, block	25	46.3
Bedroom ceiling (the roof is built with any of the following materials) ^a,b^		
Cardboard, palm, shingle (tejamanil) or wood, metal foil, foil of asbestos, does not know.	28	52.8
Concrete slab, bulkhead, brick, block	25	47.1
Living room walls (the walls are built with any of the following materials) ^a,b^		
Adobe, foil of asbestos and cardboard, does not know	15	28.3
Bulkhead, brick, block	38	71.6
Kitchen walls (the walls are built with any of the following materials) ^a,b^		
Adobe, foil of asbestos and cardboard, does not know	14	26.4
Bulkhead, brick, block	39	73.5
Bedroom walls (the walls are built with any of the following materials) ^a,b^		
Adobe, foil of asbestos and cardboard, does not know	10	19.2
Bulkhead, brick, block	42	80.7
Inside walls are painted ^a,b^	36	65.45
Outside walls are painted ^a,b^	38	69.09
Drinking water supply inside the house ^a,b^	37	68.5
Drinking water supply once a week ^a,b^	32	59.2
Washing work clothes together with family clothes ^a,b^	20	37.0
Use of pesticides at home ^a,b^	35	64.8
Separates organic and inorganic garbage ^a,b^	27	50.0
Use of toilet connected to septic tank ^a,b^	51	94.4

^a^ 26 variables considered to be included in the HSHI before the PCA; ^b^ Variables with at least one missing values.

**Table 2 ijerph-16-00295-t002:** Characteristics of the study population in Alpuyeca, Morelos.

Number of Participants (55)	
Characteristic	*N*
Gender ^a^	
Female (%)	54.5
Male (%)	45.5
Current weight (Kg), mean (S.D.) ^b^	34.9 (±10.3)
Age (Years), mean (S.D.) ^b^	9.8 (±1.4)
Body Mass Index ^c^	
Mild malnutrition (%)	5.5
Normal (%)	47.4
Obesity (%)	23.6
Overweight (%)	20.0
Breastfeeding (%) ^d^	92.5
Breastfeeding period (months) (*n* = 49) mean (S.D) ^e^	11.3 (±9.9)
Residence period (years) (*n* = 52) mean (S.D) ^e^	8.0 (±2.4)
Education of the mother (%) ^b^	
Elementary	64.1
> Other (middle school, high school, bachelor’s degree, postgraduate studies)	26.4
Education of the father (%) ^b^	
Elementary	79.2
> Other (middle school, high school, bachelor’s degree, postgraduate studies)	20.7
School ^a^	
General Ignacio Maya (%)	47.4
A.N. Urueta (%)	38.1
Andrés Aponte (%)	14.3
Smoking inside/outside the house (%) ^b^	37.0

^a^*n* = 55; ^b^
*n* = 53; ^c^ BMI was calculated according to the World Health Organization (WHO) recommended criteria for population 5 to 9 years); ^d^
*n* = 54; ^e^
*n* = 52.

**Table 3 ijerph-16-00295-t003:** Selection of dominant variables in each component for the construction of the healthy-sustainable housing index (HSHI).

Component	Load (Eigenvector)	% of Variance Explained	Dominant Variables
First component	0.3899 0.3884 0.3649	18.70	Ventilation Lighting Washing work clothes together with family clothes
Second component	0.4272 0.4268 0.3372	15.92	Construction material used in the ceiling of the living room Indoor painted walls Type of bathroom
Third component	0.5049 0.3692 0.3741	11.94	Material used for the floor of the house Construction material used in the bedroom wall Frequency of drinking water supply
Fourth component	0.4050 0.4693 0.6035	9.15	Area where houses are located Home cleaning frequency Use of pesticides at home
Fifth component	0.4251 0.4950	7.67	Drinking water supply inside the house Separating organic and inorganic garbage
Total		63.38	

**Table 4 ijerph-16-00295-t004:** Selected crude and adjusted association measures between HSHI and symptoms frequency (*n* = 55).

Caption	Crude and Adjusted Logistic Regression Models for the Prevalence of Respiratory Symptoms and Profiles						Crude and Adjusted Poisson Regression Models for the Duration of Respiratory Symptoms and Profiles					
	Crude OR	P	IC	Adjusted OR †	P	IC	Crude IRR	P	IC	Adjusted IRR †	P	IC
*Component 1*												
Dry cough	1.860	0.269	(0.619, 0.588)	3.327	0.120	(0.731, 15.136)	---	---	---	---	---	---
Ear pain	0.606	0.484	(0.149, 2.464)	0.017	0.080	(0.000, 1.633)	---	---	---	---	---	---
Hoarseness	0.435	0.280	(0.336, 2.976)	0.153	0.115	(0.015, 1.581)	---	---	---	---	---	---
Nose irritation	0.293	0.101	(0.068, 1.268)	0.071	0.072	(0.004, 1.270)	0.250	0.013*	(0.084, 0.748)	0.110	0.018	(0.018, 0.680)
URI	0.675	0.533	(0.196, 2.322)	0.259	0.143	(0.042, 1.578)	0.765	0.467	(0.371, 1.574)	0.155	0.006 *	(0.041, 0.582)
Allergy	0.188	0.049*	(0.035, 0.992)	0.002	0.075	(0.000, 1.857)	0.250	0.080	(0.053, 1.177)	0.043	0.069	(0.001, 1.277)
Runny nose	0.454	0.166	(0.149, 1.386)	0.254	0.118	(0.045, 1.418)	0.667	0.209	(0.339, 1.311)	0.382	0.049 *	(0.146, 0.997)
Sore throat	---	---	---	---	---	---	0.636	0.186	(0.326, 1.244)	0.468	0.122	(0.179, 1.224)
Sneezing	---	---	---	---	---	---	0.636	0.105	(0.368, 1.100)	0.454	0.070	(0.193, 1.066)
*Component 2*												
Common cold	0.388	0.098	(0.126, 1.192)	0.086	0.022*	(0.010, 0.699)	0.760	0.367	(0.419, 1.380)	0.342	0.021 *	(0.138, 0.848)
URI	0.675	0.533	(0.196, 2.322)	0.268	0.171	(0.041, 1.768)	1.000	1.000	(0.489, 2.046)	0.232	0.015 *	(0.072, 0.749)
Allergy	0.606	0.484	(0.149, 2.464)	0.077	0.122	(0.003, 1.991)	0.667	0.530	(0.188, 2.362)	0.175	0.151	(0.016, 1.887)
*Component 3*												
Dry cough	0.735	0.579	(0.247, 2.186)	0.311	0.120	(0.071, 1.359)	1.053	0.873	(0.562, 1.972)	0.463	0.111	(0.180, 1.192)
Runny nose	---	---	---	---	---	---	1.500	0.209	(0.797, 2.824)	1.932	0.125	(0.832, 4.485)
Sore throat	---	---	---	---	---	---	2.000	0.050*	(1.000, 3.999)	1.947	0.134	(0.814, 4.656)
Nose irritation	---	---	---	---	---	---	2.333	0.082	(0.897, 6.072)	2.230	0.209	(0.639, 7.787)
Sneezing	---	---	---	---	---	---	1.842	0.032*	(1.054, 3.220)	2.148	0.049 *	(1.004, 4.599)
*Component 4*												
Coughing up mucus	0.733	0.578	(0.246, 2.189)	0.228	0.073	(0.045, 1.147)	---	---	---	---	---	---
Nose irritation	---	---	---	---	---	---	3.000	0.033*	(1.090, 8.254)	2.284	0.153	(0.736, 7.092)
*Component 5*												
Common cold	1.364	0.578	(0.457, 4.071)	2.777	0.173	(0.639, 12.070)	---	---	---	---	---	---
Sore throat	---	---	---	---	---	---	1.400	0.320	(0.722, 2.716)	2.778	0.046 *	(1.019, 7.572)
URI	---	---	---	---	---	---	3.286	0.006*	(1.410, 7.657)	3.878	0.015 *	(1.298, 11.588)
*HSHI*												
Ear pain	0.188	0.049*	(0.035, 0.992)	0.042	0.057	(0.002, 1.095)	---	---	---	---	---	---

† Model adjusted by gender, age, breastfeeding, BMI, number of people living in the house, possession of pets, presence of pests, presence of mold, education of the father and education of the mother; * Statistically significant model (*p* < 0.05).

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
