# Peer review of "Healthy-Sustainable Housing Index: A Pilot Study to Link Architecture and Public Health in a Semi-Urban Community in Mexico"

_ijerph, 2019, doi:10.3390/ijerph16030295_

Reviewer 1 Report

This paper evaluated the link between housing and children´s respiratory symptoms, through
the construction of an index (HSHI) based on the definition of healthy-sustainable housing
criteria, in a semi-urban community from Morelos, Mexico. The topic is very interesting and
the paper is generally well written. The reviewer recommends the publication of this paper.

One minor comment: It was stated that “The main findings of this pilot study suggest living
in a house with a sufficient quality, expressed by the HSHI, lowered the frequency of
respiratory symptoms...” However, from Figure 2(b), it seems that living in a house with a
sufficient quality increased the frequency of sore throat, hoarseness, and sneezing. Please
clarify or rephrase the statements to avoid any potential confusion.

Reviewer 2 Report

Thank you for the opportunity to review this paper. It is well done and can be published
with only minor modifications as suggested below.

Introduction
Line 43. There are other definitions of a healthy house, which should be referenced here.
See Surgeon General’s call to action on healthy housing at
https://www.ncbi.nlm.nih.gov/books/NBK44192/ . Reference 2 from WHO is outdated and
should be replaced by WHO’s environmental burden of disease at
http://www.euro.who.int/en/publications/abstracts/environmental-burden-of-diseaseassociated-
with-inadequate-housing.-summary-report
2.2.2 How exactly was ventilation measured?
2.3 line 38 Consider adding the National Healthy Housing Standard at
http://nchh.org/tools-and-data/housing-code-tools/national-healthy-housing-standard/ You
may also consider referencing the healthy housing rating system in Great Britain
https://www.gov.uk/government/publications/housing-health-and-safety-rating-systemguidance-
for-landlords-and-property-related-professionals
It is not clear who collected the housing data. Were these trained housing inspectors?
Researchers? Occupants?
3.1 line 23. I think you mean “high” humidity, correct?
Table 1. Does this mean that only 63% of homes had parents? Does this mean 37% of home
with children have no parents?
p. 6 line 11. This finding of an increased risk of 5.3 times more allergies is a main finding
and should be presented in the abstract
p. 7, line 2. This statement appears to be incorrect. Does living in a poor quality home
actually reduce colds or increase them?

end

Round  2

Reviewer 1 Report

The authors have addressed my comments.

Reviewer 2 Report

This second revision can be published with no additional changes, the authors have responded to previous reviewer comments adequately.